# Parameterization of Light Absorption of Phytoplankton, Non-Algal Particles and Coloured Dissolved Organic Matter in the Atlantic Region of the Southern Ocean (Austral Summer of 2020)

**Tatiana Churilova** [1,*] **, Natalia Moiseeva** [1] **, Elena Skorokhod** [1] **, Tatiana Efimova** [1] **, Anatoly Buchelnikov** [1,2] **, Vladimir Artemiev** [3] **and Pavel Salyuk** [4]

[1] A.O. Kovalevsky Institute of Biology of the Southern Seas of the Russian Academy of Science, 299011 Sevastopol, Russia
[2] Laboratory of Molecular and Cellular Biophysics, Sevastopol State University, 299053 Sevastopol, Russia
[3] Shirshov Institute of Oceanology, Russian Academy of Sciences, 117997 Moscow, Russia
[4] V.I. Ilichev Pacific Oceanological Institute of the Far Eastern Branch of the Russian Academy of Science, 690041 Vladivostok, Russia
[*] Correspondence: tanya.churilova@ibss-ras.ru

**Abstract:** Climate affects the characteristics of the Southern Ocean ecosystem, including bio-optical properties. Remote sensing is a suitable approach for monitoring a rapidly changing ecosystem. Correct remote assessment can be implemented based on a regional satellite algorithm, which requires parameterization of light absorption by all optically active components. The aim of this study is to analyse variability in total chlorophyll $a$ concentration (TChl-$a$), light absorption by phytoplankton, non-algal particles (NAP), coloured dissolved organic matter (CDOM), and coloured detrital matter (CDM = CDOM+NAP), to parameterize absorption by all components. Bio-optical properties were measured in the austral summer of 2020 according to NASA Protocols (2018). High variability (1–2 orders of magnitude) in TChl-$a$, absorption of phytoplankton, NAP, CDOM, and CDM was revealed. High variability in both CDOM absorption (uncorrelated with TChl-$a$) and CDOM share in total non-water absorption, resulting in a shift from phytoplankton to CDOM dominance, caused approximately twofold chlorophyll underestimation by global bio-optical algorithms. The light absorption of phytoplankton (for the visible domain in 1 nm steps), NAP, CDOM, and CDM were parametrized. Relationships between the spectral slope coefficient ($S_{CDOM}/S_{CDM}$) and CDOM (CDM) absorption were revealed. These results can be useful for the development of regional algorithms for Chl-$a$, CDM, and CDOM monitoring in the Southern Ocean.

**Keywords:** chlorophyll $a$; light absorption; phytoplankton; non-algal particles; coloured dissolved organic matter; remote sensing; Southern Ocean

## 1. Introduction

The Southern Ocean is identified as a very productive region of the global ocean [1], providing important ecosystem services, particularly fisheries [2,3] and maintenance of biodiversity [1,4]. However, the Southern Ocean is currently the most vulnerable area to ongoing atmospheric and oceanic warming [5–8]. Climate affects the biological and physical characteristics of the Southern Ocean ecosystem [1]. Ocean warming leads to glacial and ice sheet melting [1]. Ice melting is associated with the release of suspended dissolved organic matter (including its coloured fraction—CDOM) into the surrounding waters [9]. The ice melting opens new ice-free areas for additional primary production (PP) [1,10]. In the Southern Ocean, despite high macronutrient concentration [11,12], low surface chlorophyll $a$ concentration (mean is ca. 0.25 mg m$^{-3}$) is typical for a major part (ca. 48%) of the ocean [12]. It is related to the strong limitation placed upon phytoplankton growth and, consequently, chlorophyll $a$ concentration by the lack of ambient light due to deep vertical

mixing, persistent cloud cover, large solar zenith angles in polar regions and particularly in the Southern Ocean [13,14]. Moreover, the phytoplankton growth in the Southern Ocean is limited significantly by micronutrients, in particular by iron availability, which changes due to ice melting [15–17]. All these factors influence the inherent optical properties (IOPs) and biological productivity of the Southern Ocean [10]. Due to rapid climatic-driven changes in the physical environments, the assessment of the current state of water quality and ecosystem productivity becomes especially relevant. Since phytoplankton is the basis of marine food webs and is the most sensitive and quick-responding to changes in the environment, the biomass and species structure of phytoplankton and its photosynthetic characteristics are considered as key indicators of the state and productivity of the Antarctic marine ecosystems [1,7]. The concentration of the main photosynthetically active pigment, chlorophyll *a*, is widely used as a marker of phytoplankton biomass [18,19]. In addition to *in situ* observations, long-term (multi-decadal) remote sensing data series can be used to obtain fundamental observational data of upper water layer properties with a high spatial and temporal resolution, which are crucial for understanding the current state of the Southern Ocean ecosystem and investigation of climate-scale phenomena as well as predicting possible changes [1]. Evaluations of the accuracy of remote sensing of chlorophyll *a* concentration (Chl-*a*) showed good performance ($\pm$35%) in the open ocean [20]. However, global satellite-based Chl-*a* algorithms were shown to perform poorly in the Southern Ocean, resulting in underestimations of Chl-*a* up to almost twice in most cases [21–24]. Several regional Chl-*a* algorithms were proposed for the Southern Ocean to improve the accuracy of satellite Chl-*a* retrievals [20,25]. However, a recent assessment of the accuracy of these algorithms [24] showed that the application of the algorithms led to the underestimation of Chl-*a*. In this recent work [24], the performance of both global and regional satellite algorithms (currently available for the Western Antarctic Peninsula) was assessed based on the most comprehensive in situ dataset ever collected from the Western Antarctic Peninsula (N = 1812, with Chl-*a* concentration from 0.01 mg m$^{-3}$ to 46.58 mg m$^{-3}$). Noted [21–23] differences between in situ and remote sensing Chl-*a* data can be large due to regional differences in Chl-*a* specific phytoplankton light absorption coefficient, dominating phytoplankton taxon and low particulate backscattering [20,21,26]. In the Indian sector of the Southern Ocean, similarly to the Atlantic sector, the underestimation of Chl-*a* based on satellite data was associated with the pigment package effect and the structure of phytoplankton community, which affected Chl-*a* specific phytoplankton light absorption coefficient [14,27,28]. In a recent study in the Western Antarctic Peninsula region [24], unlike previous ones, no clear link was revealed between Chl-*a* underestimation and the impact of pigment packaging on Chl-*a* specific phytoplankton light absorption, nor with the species composition of phytoplankton. This indirectly indicates the influence of other optically active components on the accuracy of the algorithms. The signal "visible" by satellite scanners (remote sensing reflectance, R$_{rs}$) is determined in general by scattering and absorption of light by all in-water optically active substances. The absorption has a predominant effect on R$_{rs}$ compared to scattering [29,30]. The total absorption ($a(\lambda)$) consists of the absorption by all optically active components [30]:

$$a(\lambda) = a_{ph}(\lambda) + a_{NAP}(\lambda) + a_{CDOM}(\lambda) + a_w(\lambda), \tag{1}$$

$$a_p(\lambda) = a_{ph}(\lambda) + a_{NAP}(\lambda), \tag{2}$$

where $a_{ph}(\lambda)$, $a_{NAP}(\lambda)$, $a_{CDOM}(\lambda)$, $a_p(\lambda)$ and $a_w(\lambda)$ are light absorption coefficients at the wavelength ($\lambda$) of phytoplankton, non-algal particles (NAP), coloured dissolved organic matter (CDOM), particles, and pure water, respectively. The IOPs, particularly $a_{ph}(\lambda)$, $a_{NAP}(\lambda)$ and $a_{CDOM}(\lambda)$, largely determine the variability in the spectral features of R$_{rs}$. In general, not only phytoplankton but also CDOM and NAP absorption can affect the R$_{rs}$ signal. The CDOM content was found to be highly variable in the Southern Ocean due to both physical (ice melting, photobleaching) and biological factors (phytoplankton, bacteria, and other pelagic organisms' production) [1,6,7]. It was shown that the NAP concentration

in the surface waters strongly depended on ice melting [9]. The variability of spectral absorption by phytoplankton was studied in the Southern Ocean [14,26,27,31,32]. However, only single studies of spectral absorption by all optical components (phytoplankton, NAP, and CDOM) are known to date [28,33].

Recently, a new satellite algorithm was developed for more accurate Chl-*a* estimation [20]. For a comprehensive assessment of the quality and productivity of this climate-sensitive region, additional indicators could be provided by a three-bands algorithm dividing total absorption into two, viz., absorption by phytoplankton $a_{ph}(\lambda_r)$ and coloured detrital matter ($a_{CDM}(\lambda_r) = a_{CDOM}(\lambda_r) + a_{NAP}(\lambda_r)$) [34]. This three-bands algorithm was developed for the Black Sea based on regional spectral bio-optical properties, which allowed the retrieval of Chl-*a* based on the revealed phytoplankton absorption parameterization [35]. In addition to Chl-*a*, this algorithm provides two indicators: (1) $a_{ph}(\lambda)$, i.e., the phytoplankton photosynthetic characteristic required for PP assessment by the spectral approach; (2) $a_{CDM}(\lambda_r)$ used as a marker of CDOM and NAP content and required for the spectral downwelling radiance model enabled for the full spectral PP model [36]. The accuracy of the three-bands algorithm was proven by validation, comparing it to other algorithms known for the Black Sea, including the standard algorithm [37]. Adaptation of the three-bands algorithm for the Southern Ocean requires parametrization of the absorption by all optical components based on newly collected in situ data. It should be noted that for different scanners (to obtain a merged product) as well as for another model (light and PP), the relationship between $a_{ph}(\lambda)$ and Chl-*a* is required to be parameterized for the entire visible domain, which has not been done before. In the Southern Ocean, the relationship between $a_{ph}(\lambda)$ and Chl-*a* (summed with phaeopigments) was previously obtained only for particular wavelengths corresponding to the maxima of the $a_{ph}(\lambda)$ spectrum [26,32].

The aim of this study is to analyse variability in chlorophyll *a* concentration, and spectral light absorption coefficients by phytoplankton, NAP, CDOM, and CDM, to parameterize the absorption by all components (including the relationship between $a_{ph}(\lambda)$ and the sum of Chl-*a* with phaeopigments for the visible domain in 1 nm steps) based on newly (11 January–4 February 2020) collected in situ data in the Atlantic region of the Southern Ocean (Bransfield Strait, Falkland Current, Drake Passage, and the Powell Basin). One of the objectives of the study was to evaluate the accuracy of the standard satellite product, Chl-*a*, based on a comparison with the measurement results.

## 2. Materials and Methods

### 2.1. Sampling

Data were collected during a scientific cruise on RV "Akademik Mstislav Keldysh" in the Atlantic region of the Southern Ocean in the austral summer of 2020 (11 January–4 February). The study area covered the Bransfield Strait (BS), Falkland (Malvinas) Current (FC), Drake Passage (DP), and the Powell Basin (PB) (Figure 1). Salinity and temperature profiles were recorded with a Sea-Bird SBE-911 plus CTD unit. The chlorophyll *a* fluorescence profiles were recorded with the fluorometer sensor Minitracka-II (Chelsea), which was installed on the transparency sensor PUM-200. Photosynthetically available radiation (PAR) incident on the sea surface was measured with Li-COR set on the deck (LI–190SA). PAR vertical profile was measured with a submersible sensor (LI-192). Water samples were collected at 3–7 depths using Niskin bottles. The sampling depths were chosen based on the temperature, salinity, PAR, and chlorophyll *a* fluorescence profiles. Immediately after sampling, water aliquots were filtered gently (under low vacuum) under red light for analyses of Chl-*a* and for light absorption by particles, phytoplankton, NAP, and CDOM.

### 2.2. Chlorophyll a Concentration

The concentration of chlorophyll *a* (Chl-*a*) and phaeopigments was determined spectrophotometrically with 90% acetone using Whatman GF/F glass fiber filters (25 mm diameter, 0.7 μm size pore) [38,39].

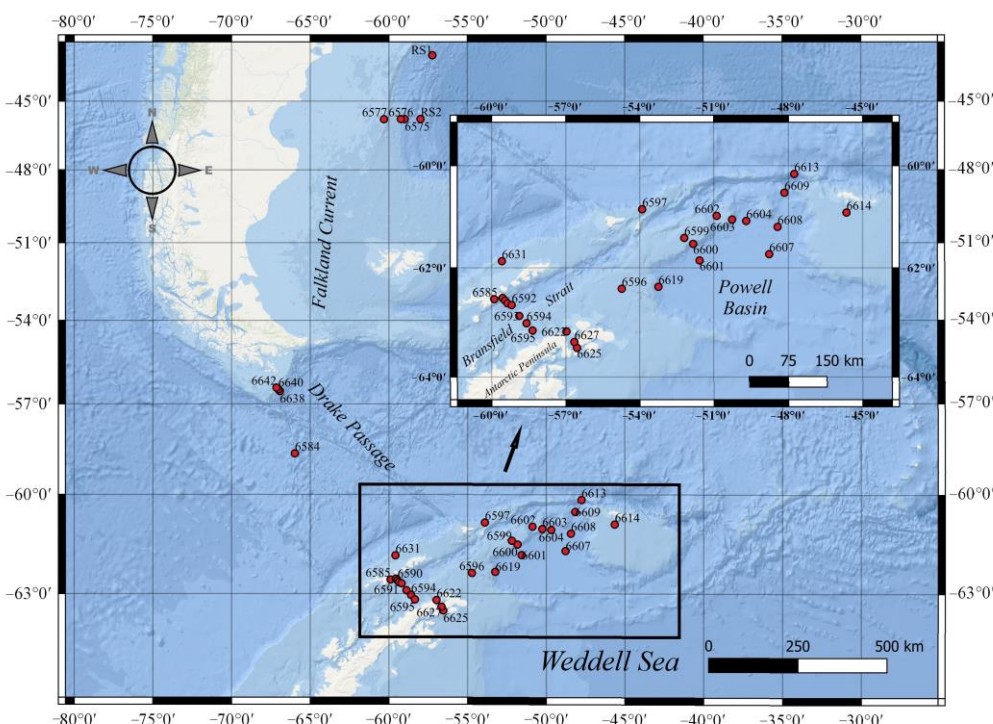

**Figure 1.** Station map showing station locations (black circles) covered during cruise No. 79 of RV "Akademik Mstislav Keldysh" from 11 January–4 February 2020.

Satellite-derived Chl-*a* (chlor_a) were extracted from the "Ocean Color Web" portal [40]. The chlor_a are level 3 data with a spatial resolution of approximately 4 × 4 (pixel) km, retrieved based on colour scanners Aqua-MODIS, SNPP-VIIRS, and S3A-OLCI measurements and by merging various ocean colour scanners measurements (Merged_ATV dataset). The default algorithm for chlor_a estimation [41] employs the standard OC3/OC4 (OCx) band ratio algorithm [42] merged with the colour index [43]. The satellite data were selected for the same day of in situ measurements from an area of 3 × 3 pixels around each station of in situ measurements. Then chlor_a values were filtered by outliers by the 1.5 IQR (Inter Quartile Range) rule. Data left after the filtering were averaged.

Statistical analysis of comparison in situ and satellite data of Chl-*a* was carried out based on metrics—systemic bias (Equation (3)) and Median Absolute Error (MAE) (Equation (4)) calculated according to refs. [44,45]:

$$bias = 10 \left( \frac{1}{N} \sum_{i=1}^{N} (log_{10}(Chl_{sat}(i)) - log_{10}(Chl_m(i))) \right), \quad (3)$$

$$MAE = 10 \left( \frac{1}{N} \sum_{i=1}^{N} |log_{10}(Chl_{sat}(i)) - log_{10}(Chl_m(i))| \right) \quad (4)$$

### 2.3. Particulate Absorption

Particulate absorption was measured by the quantitative filter technique (QFT) using wet filters in accordance with the current NASA protocol [46]. Water samples (1–2 L) were filtered on Whatman GF/F glass fiber filters (25 mm diameter, 0.7 μm size pore), then filters were wrapped in aluminum foil (chlorophyll *a*) and put in plastic boxes (particulate absorption), which were stored in liquid nitrogen until analysis. Optical density (OD, dimensionless) of total particulate matter ($OD_p(\lambda)$, dimensionless) was measured by scanning the sample filter from 350 to 750 nm in 1 nm increments using a dual-beam UV/Vis spectrometer Lambda 35 (Perkin Elmer) equipped with an integrating sphere. To separate the phytoplankton pigments from non-algal particles (NAP) within the particulate matter, the pigments were extracted with hot methanol [47]. To remove phycobilin pigments,

subsequent extraction with hot water was used [48]. After depigmentation, the sample was scanned again to obtain OD of NAP ($OD_{NAP}(\lambda)$, dimensionless). To minimize the differences between the sample and blank filters, all the sample spectra were shifted to zero near the infrared region by subtracting the average absorbance from 715 to 750 nm. The OD were corrected for path length amplification using the β-correction algorithm [49]. The OD corrected for pathlength amplification ($OD_{p/corr}(\lambda)$ or $OD_{NAP/corr}(\lambda)$) were converted to absorption coefficients $a_p(\lambda)$ (in $m^{-1}$) or $a_{NAP}(\lambda)$ (in $m^{-1}$):

$$a_j(\lambda) = 2.303 \times OD_{j/corr}(\lambda)/(V/S).\tag{5}$$

where $j$ denotes total particles ($p$) or NAP, 2.303 = ln 10, $V$ is the water filtration volume (in $m^3$), and $S$ is the filter clearance area (in $m^2$).

Phytoplankton component $a_{ph}(\lambda)$ (in $m^{-1}$) of total particulate matter was then determined by subtracting $a_{NAP}(\lambda)$ from $a_p(\lambda)$:

$$a_{ph}(\lambda) = a_p(\lambda) - a_{NAP}(\lambda).\tag{6}$$

Chlorophyll *a* specific absorption coefficient of phytoplankton ($a_{ph}^*(\lambda)$, $m^2\ mg^{-1}$) was calculated by dividing $a_{ph}(\lambda)$ by the sum of chlorophyll *a* and phaeopigment concentration (TChl-*a*). Regression analysis was conducted to determine the relationship between $a_{ph}(\lambda)$ and TChl-*a* from 400 to 700 nm in 1 nm increments [50]:

$$a_{ph}(\lambda) = A(\lambda) \times TChl\text{-}a^{B(\lambda)},\tag{7}$$

where $A(\lambda)$ and $B(\lambda)$ are the coefficients of non-linear fit derived from our data.

Statistical analysis of the similarity (or not) of the $A(\lambda)$ and $B(\lambda)$ at 438 and 678 nm within the euphotic layer was executed based on the Fisher criterion.

### 2.4. CDOM Absorption

CDOM absorption was measured following the NASA protocol [46]. Water samples were filtered under low vacuum using 0.2 μm nylon filters (Sartorius Nuclepore) and GF/F (Whatman) filters for prefiltration. The filters were rinsed with deionized water (ca. 50 mL) before filtration. OD of the CDOM was measured from 250 to 750 nm in 1 nm increments using a dual-beam UV/Vis spectrometer Lambda 35 (Perkin Elmer), a 10 cm quartz cuvette, and deionized water as a reference. The $OD_{CDOM}(\lambda)$ was null-point corrected at 700 nm by subtraction of the mean value over a 5 nm interval around 700 nm from each wavelength [51]. The CDOM absorption coefficient ($a_{CDOM}(\lambda)$, $m^{-1}$) was calculated from the null corrected OD:

$$a_{CDOM}(\lambda) = (OD_{CDOM}(\lambda) - OD_{deion/w}(\lambda)) \times 2.303/l,\tag{8}$$

where $l$ is the cuvette path length (in meters).

The $a_{CDOM}(\lambda)$, $a_{NAP}(\lambda)$, and $a_{CDM}(\lambda)$ (CDM = NAP + CDOM) spectra can be represented as exponential functions as follows [52]:

$$a_i(\lambda) = a_i(\lambda_r) \times e^{-S_i \cdot (\lambda - \lambda_r)},\tag{9}$$

where $i$ denotes NAP, CDOM, or CDM, the wavelength $\lambda_r$ is the reference wavelength, in this study $\lambda_r$ = 438 nm, and $S_i$ denotes the spectral slope. $S_{NAP}$ and $S_{CDOM}$ ($S_{CDM}$) were obtained by means of a non-linear least square fit of an exponential function (Equation (4)) to the data from 400 to 700 nm and from 350 to 500 nm [46], respectively. The calculation routine consists in finding the smallest value of the quadratic discrepancy function Δ,

i.e., the sum of squares of differences of the measured and theoretically calculated values of $a_i(\lambda)$ at each wavelength, by fitting the guess parameters $a_i(438)$ and $s_i$:

$$\Delta = \sum_{\lambda=400}^{700} \left( a_i^{\text{exper}}(\lambda) - a_i^{\text{theor}}(\lambda) \right)^2. \tag{10}$$

To assess the quality of fitting, we used the coefficient of determination

$$r^2 = 1 - \frac{\Delta}{\sum_{\lambda=400}^{700} \left( a_i^{\text{exper}}(\lambda) - \left\langle a_i^{\text{exper}}(\lambda) \right\rangle \right)^2}, \tag{11}$$

where $\left\langle a_i^{\text{exper}}(\lambda) \right\rangle$ is the average of experimental spectrum $a_i(\lambda)$ over all wavelengths.

The computational algorithm was implemented using Python 3 programming language.

To justify statistically the possibility of using the same coefficient of Equation (7) of phytoplankton absorption within the euphotic zone, Fisher's and Student's criteria were used.

The euphotic zone ($Z_{eu}$) was defined as the depth where the photosynthetically available radiation was 1% of that incident at the surface.

## 3. Results

### 3.1. Hydrophysical Characteristics

In the area studied, hydrographical characteristics were different (Figure 2). The surface temperature varied from $-0.47\,^\circ\text{C}$ to $3.19\,^\circ\text{C}$ and was $1.29 \pm 1.4\,^\circ\text{C}$ on average in the BS. In the PB, the surface temperature was lower than in the BS; it varied from $-0.77\,^\circ\text{C}$ to $2.01\,^\circ\text{C}$ and was $0.72 \pm 0.89\,^\circ\text{C}$ on average. Water in the FC was warmer than in other investigated regions: the surface temperature was in the range of $10.6$–$13.0\,^\circ\text{C}$. In the DP, the temperature at the surface was $7.4\,^\circ\text{C}$ (the bio-optical investigation was carried out at one station). At all stations, the vertical temperature distribution had low gradients ($0.001$–$0.78\,^\circ\text{C m}^{-1}$), although, in the layer of $0$–$120$ m, the temperature decreased to $6$–$7\,^\circ\text{C}$ in the FC, in $1.4$–$3\,^\circ\text{C}$ in the DP and the BS, and in $0.7$–$3.0\,^\circ\text{C}$ in the PB (Figure 2).

The salinity of the surface layer varied from 31.2 to 35.8‰ and was $33.6 \pm 1.0$‰ on average. In the salinity profile, there was a minimum in the layer from 0 to ca. 25 m. In this minimum, the salinity was 0.5–5‰ lower than the salinity in the surface layer. In a deeper water layer (more than ca. 25 m), salinity remained constant at all stations at the level of ca. 34.5‰ (Figure 2).

The $Z_{eu}$ varied about three times from 29 m in the PB (station 6609) to 87 m in the DS (station 6584) and was, on average, $60 \pm 15$ m. At stations performed at night, $Z_{eu}$ was estimated by an empirical relationship derived from measured $Z_{eu}$ and total non-water absorption at 438 nm ($a_{tot}(438)$), measured at surface (1 m depth) for the stations performed during daylight ($Z_{eu} = 27 \times a_{tot}(438)^{-0.344}$, $n = 18$, $r^2 = 0.51$).

### 3.2. Chlorophyll a Concentration

Variability in the profiles of hydrographical characteristics resulted in the different vertical distributions of chlorophyll *a* fluorescence (F). At some stations, fluorescence intensity was uniformly distributed within the euphotic layer (Figure 2, stations 6584, 6599, 6619, 6627). Other stations demonstrated depth-dependent variability in F. Maximum of F was observed near the bottom of the euphotic layer (Figure 2, stations 6576, 6591, 6592, 6607).

TChl-*a* in the surface layer varied significantly (from 0.20 to 4.4 mg m$^{-3}$). Minimum values (0.20–0.29 mg m$^{-3}$) were measured in the DP and the deep-water region of the BS (stations 6594, 6595). Maxima TChl-*a* were obtained in the PB: 4.4 mg m$^{-3}$ at station 6609 and 2.2 mg m$^{-3}$ at station 6613. The mean surface TChl-*a* was equal to $0.72 \pm 0.35$ mg m$^{-3}$ for all stations except for 6609 and 6613 stations. Within the investigated water layer, TChl-*a* varied from 0.15 to 1.81 mg m$^{-3}$ at all stations except for two more trophic stations (6609

and 6613). In the layer of TChl-*a* maximum observed at some stations, the values of the TChl-*a* exceeded the surface ones by 1.3–1.5 times.

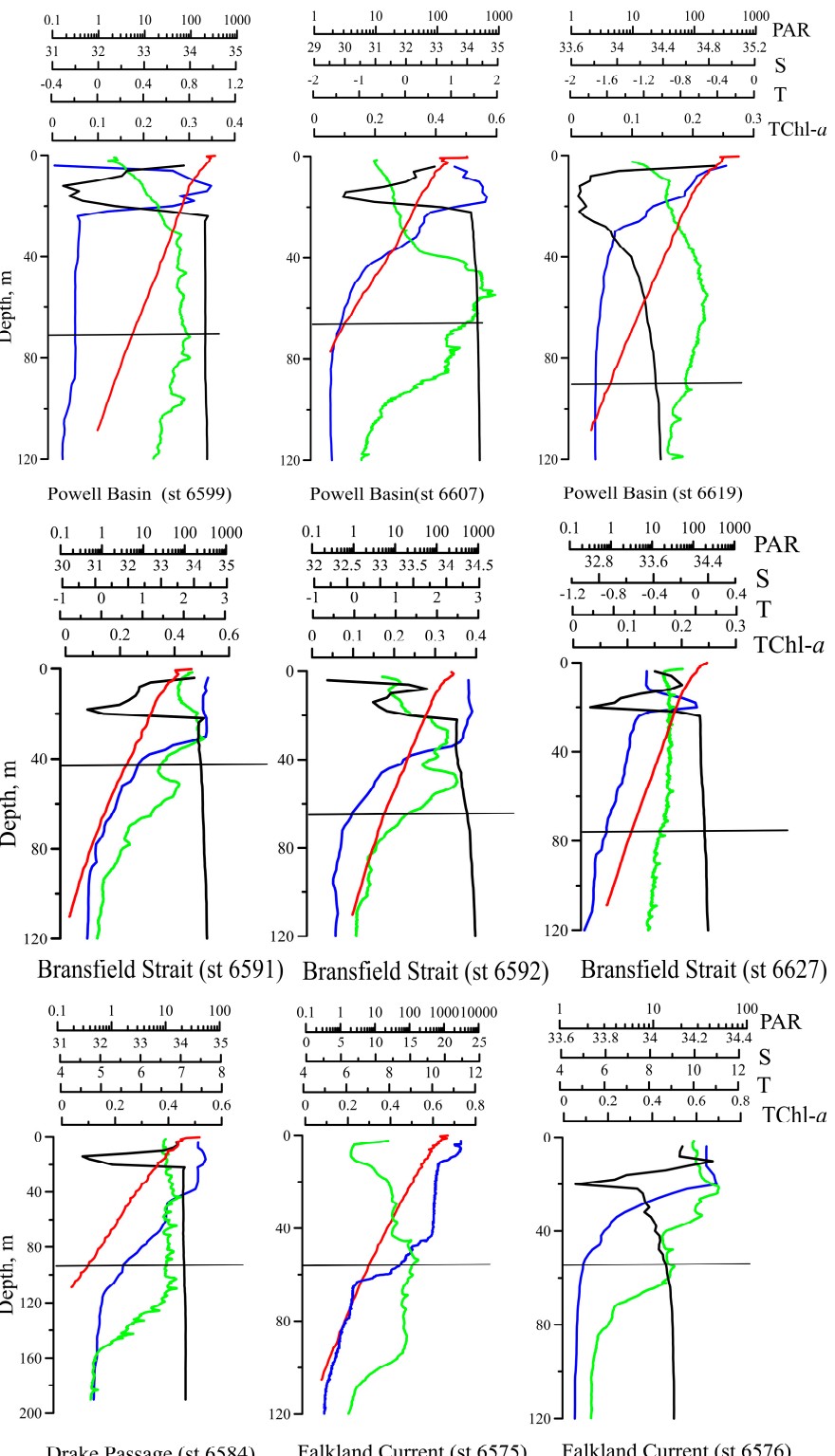

**Figure 2.** Vertical profiles of temperature (T, °C, blue line), salinity (S, ‰, black line), the fluorescence of chlorophyll *a* (TChl-*a*, mg m$^{-3}$, green line), and photosynthetic available radiance (PAR, μE m$^{-2}$ s$^{-1}$, red line) at some stations in all regions investigated in Atlantic sector of the Southern Ocean. The horizontal black line denotes the depth of 1% PAR.

### 3.3. Light Absorption by Phytoplankton

High variability in TChl-*a* was accompanied by significant (about two orders of magnitude) changes in $a_{ph}(\lambda)$ coefficients. The spectrum $a_{ph}(\lambda)$ had two main maxima typical for phytoplankton pigment complex: in the blue spectral domain at ca. 438 nm and in the red domain at ca. 678 nm ($a_{ph}(678)$) (Figure 3). The values of $a_{ph}(438)$ and $a_{ph}(678)$ varied from 0.0051 to 0.29 m$^{-1}$ and from 0.0032 to 0.12 m$^{-1}$, respectively. The maximum values were recorded in relatively high trophic waters (TChl-*a* was 2.2–4.4 mg m$^{-3}$) fixed in the PB (stations 6609 and 6613). The shape of the spectra was the same at all stations except for some stations, where TChl-*a* was more than 1 mg m$^{-3}$. At these stations in the upper layer, the shape of $a_{ph}(\lambda)$ differed from other spectra by two local maxima at wavelengths of ca. 490 and 545 nm, which were most pronounced at the 6609 station (Figure 3).

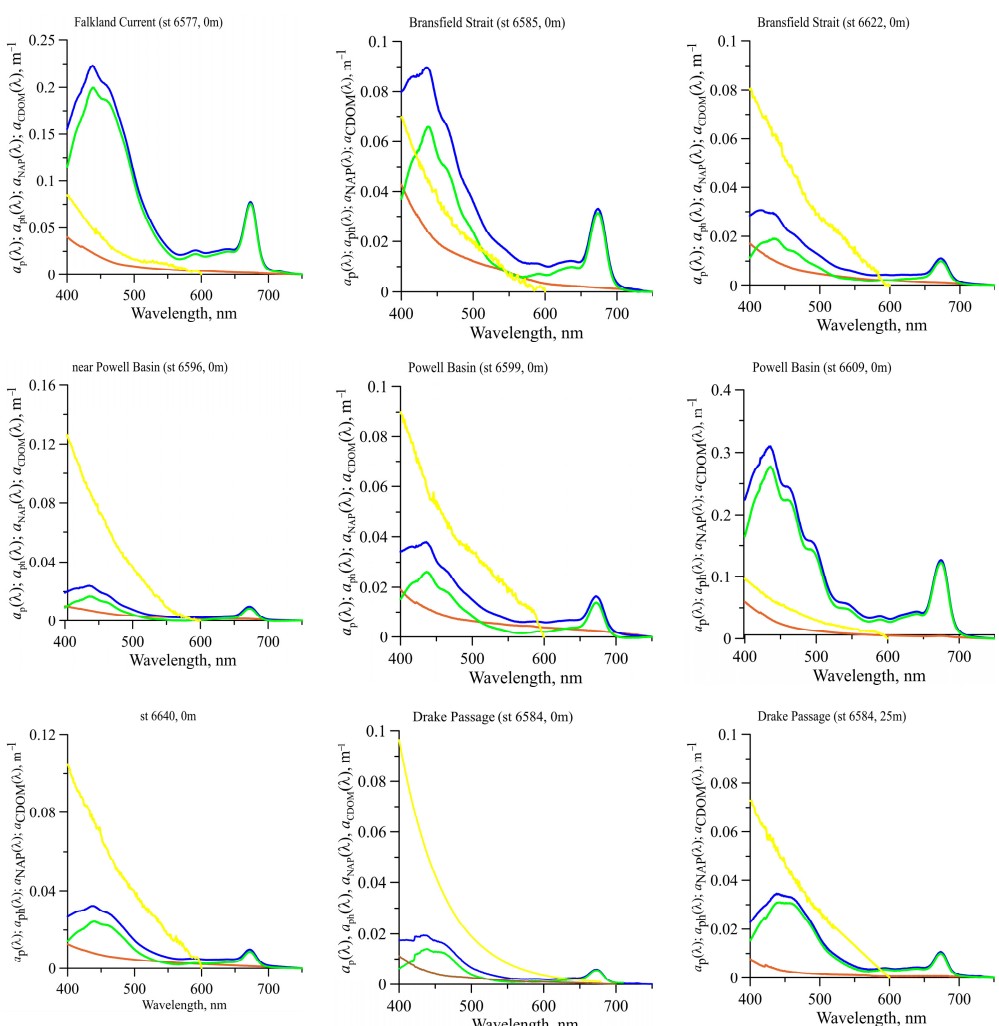

**Figure 3.** The spectral light absorption coefficient of particles ($a_p(\lambda)$, blue line), phytoplankton ($a_{ph}(\lambda)$, green line), non-algal particles ($a_{NAP}(\lambda)$, brown line) and coloured dissolved organic matter ($a_{CDOM}(\lambda)$, yellow line) at particular stations corresponding to all regions investigated in the Southern Ocean.

The local maxima correspond to the absorption bands of phycobilin pigments [53]. In the surface and the euphotic layers, the chlorophyll *a* specific absorption coefficient of 438 nm, $a_{ph}^*(438)$ ranged from 0.025 to 0.075 m$^2$ mg$^{-1}$ (with mean 0.048 ± 0.012 m$^2$ mg$^{-1}$) and from 0.021 to 0.080 m$^2$ mg$^{-1}$ (with mean 0.046 ± 0.013 m$^2$ mg$^{-1}$), respectively (Figure 4). The $a_{ph}^*(678)$ and $a_{ph}^*(438)$ varied from 0.015 to 0.028 and from 0.013 to 0.031 m$^2$ mg$^{-1}$ in the surface

and the euphotic layers, with averages equal to 0.022 ± 0.0041 and 0.022 ± 0.0048 m² mg⁻¹, respectively. The ratio between the $a^*_{ph}(438)$ and $a^*_{ph}(678)$ (R) was in the range of 1.5–2.8, with a mean for the euphotic zone of 2.1 ± 0.34 (Figure 4). At most stations, the R-value was almost constant (around 2.1) within the euphotic zone. At stations with a deep maximum of chlorophyll (four such stations out of a total number of thirty-seven stations), R decreased with depth from 2.4–2.8 in the surface layer to 1.5–1.9 near the bottom of the euphotic zone (Figure 4).

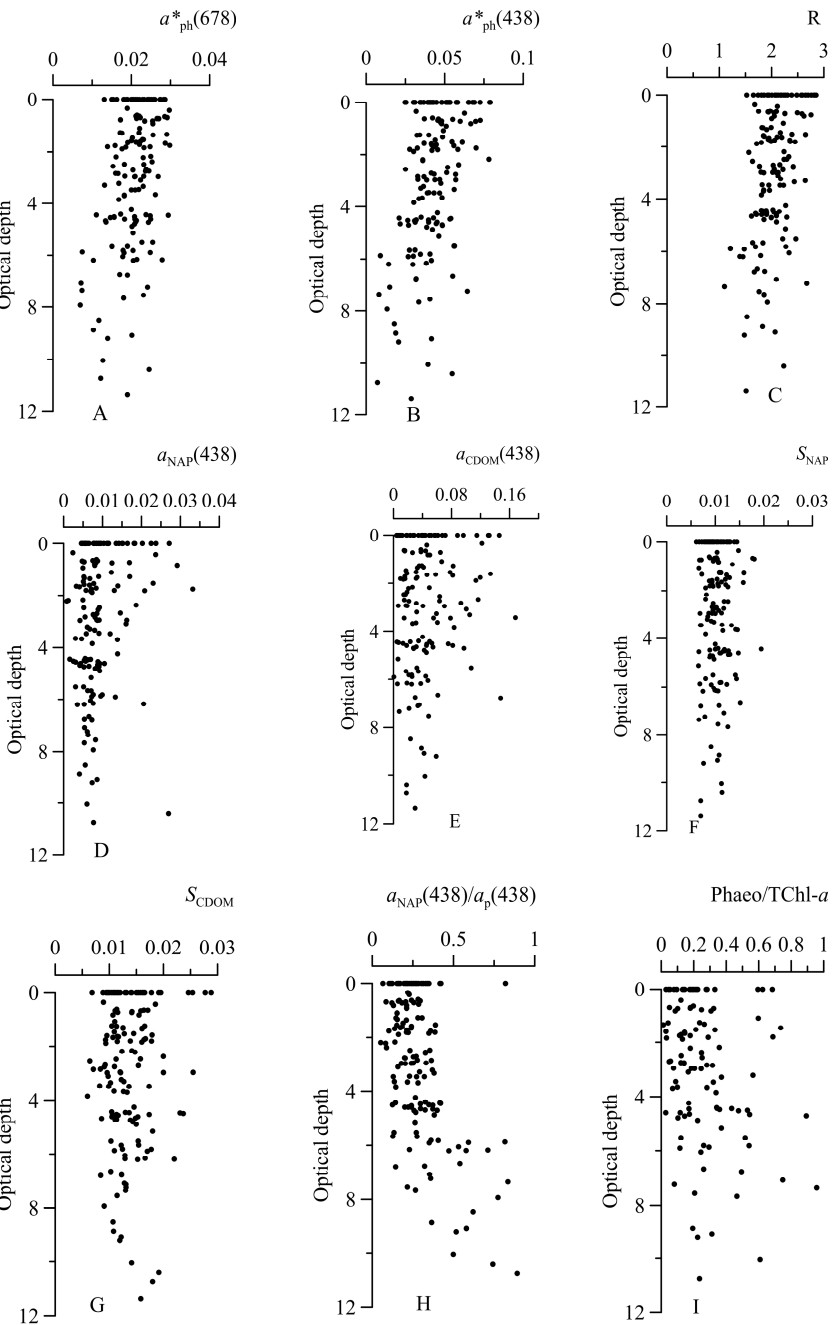

**Figure 4.** Vertical distribution of the chlorophyll *a* specific light absorption coefficient of phytoplankton at 678 nm ($a^*_{ph}(678)$) (**A**) and at 438 nm ($a^*_{ph}(438)$ ) (**B**), the ratio between $a^*_{ph}(438)/a^*_{ph}(678)$ (R) (**C**), light absorption by non-algal particles at 438 nm ($a_{NAP}(438)$ ) (**D**) and colored dissolved organic matter at 438 nm ($a_{CDOM}(438)$ ) (**E**), the spectral slope of NAP ($S_{NAP}$) (**F**) and of CDOM ($S_{CDOM}$) (**G**), the ratio between $a_{NAP}(438)$ and particulate absorption ($a_{NAP}(438)/a_p(438)$ ) (**H**) and ration between phaeopigments and its sum with chlorophyll a concentration (**I**) with optical depth.

Mostly $a_{ph}^*(438)$ and $a_{ph}^*(678)$ values did not differ in the surface layer and within the euphotic zone (Figure 4). For the surface and the layer at the end of the photosynthesis zone (10–1% PAR), the dependences of $a_{ph}(\lambda)$ on TChl-a at 438 and 678 nm were obtained. Based on Fisher's and Student's criteria [54], it was found (for 95% reliability) that they are not statistically distinguishable. Therefore, a relationship between $a_{ph}(438)$ (or $a_{ph}(678)$) and TChl-*a* was determined on the dataset obtained in the water layer above 1 % PAR in the form of a power law (see Equation (7)) (Figure 5). The obtained high values of the coefficients of determination (Table 1) demonstrate a tight relationship between $a_{ph}(438)$ ($a_{ph}(678)$) and TChl-*a* for all data retrieved within the euphotic zone. It indicates the homogeneity of the photosynthesis zone in terms of the light-absorbing characteristics of phytoplankton, namely, in terms of $a_{ph}^*(\lambda)$. The obtained coefficient B indicates no decrease in $a_{ph}^*(\lambda)$ with the growth of TChl-*a* increasing by more than an order of magnitude.

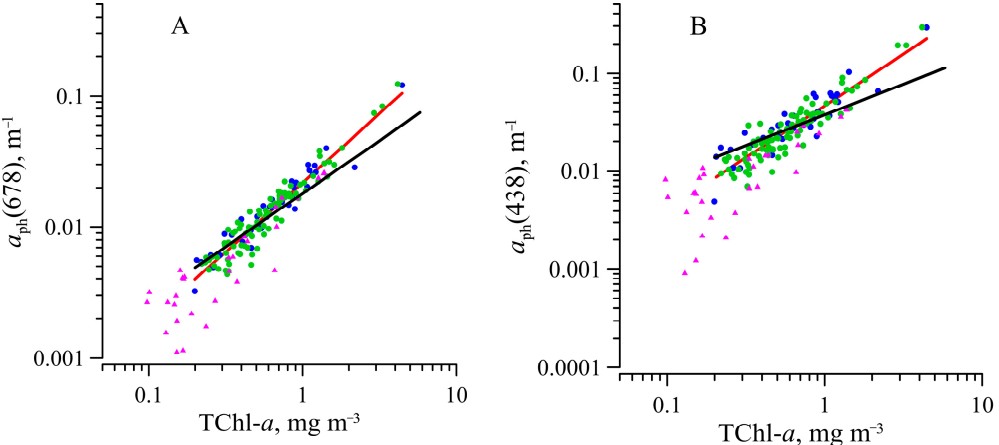

**Figure 5.** Variations of the light absorption coefficients of phytoplankton at 678 nm ($a_{ph}(678)$) (**A**) and at 438 nm ($a_{ph}(438)$ ) (**B**) as a function of the sum chlorophyll *a* and phaeopigments concentration (TChl-*a*), for various water layers: surface—blue symbols; euphotic zone, but below 1 m depth—green symbols; layer below euphotic zone—purple symbols. The red and black lines are the regression lines corresponding to Equation (7) for our data in the euphotic zone and obtained by [50], respectively. Coefficients for the regression and the coefficients of determination ($r^2$) are shown in Table 1.

**Table 1.** Coefficients, number of samples (*n*), and coefficient of determination ($r^2$) for the power fit Equation (7) described a link between phytoplankton absorption coefficient ($a_{ph}(\lambda)$) and the sum of chlorophyll *a* and phaeopigments (TChl-*a*) for data within the euphotic zone, $\lambda$ is either 438 nm or 676 nm.

| A($\lambda$) | B($\lambda$) | $r^2$ | n | Region | Reference |
|---|---|---|---|---|---|
| | | | | $\lambda$ = 438 nm | |
| 0.042 | 0.93 | 0.83 | 126 | Bransfield Strait, Falkland Current, Drake Passage, and the Powell Basin | This research |
| 0.038 | 0.63 | 0.90 | 815 | Different regions of the global ocean, not including the Southern Ocean | Bricaud et al., 1995 |
| 0.042 | 0.74 | 0.90 | 455 | Gerlache Strait, the Bransfield Strait, and the northwestern Weddell Sea | Ferreira et al., 2017 |
| | | | | $\lambda$ = 678 nm | |
| 0.018 | 0.92 | 0.92 | 126 | Bransfield Strait, Falkland Current, Drake Passage, and the Powell Basin | This research |
| 0.017 | 0.82 | 0.94 | 815 | Different regions of the global ocean, not including the Southern Ocean | Bricaud et al., 1995 |
| 0.020 | 0.79 | 0.93 | 455 | Gerlache Strait, the Bransfield Strait, and the northwestern Weddell Sea | Ferreira et al., 2017 |

The relationship (B95) obtained based on datasets from different regions of the global ocean, not including the Southern Ocean [50], is shown in Figure 5. It is evident that our data agree with B95 only in the middle of the TChl-*a* range of variability (0.5–0.6 mg m$^{-3}$). However, under lower and higher TChl-*a* values, there are deviations of our data (line correlation) from B95's line.

Comparison of the obtained coefficients (Table 1) with those for a large dataset of ocean data [50] and the Southern Ocean [26] showed the difference in B($\lambda$) coefficients. The obtained values of coefficient B (near 1) indicate no decrease in $a_{ph}^*(\lambda)$ with a rise of the TChl-*a* by more than an order of magnitude in the surface layer.

*3.4. Parameterization of a Link between Phytoplankton Absorption Coefficient and Sum of Chlorophyll a and Phaeopigments Concentration*

Good correlation between $a_{ph}(438)$ ($a_{ph}(678)$) and TChl-*a* obtained for the euphotic zone (Figure 5, Table 1) suggests that it would be appropriate to parameterize $a_{ph}(\lambda)$ with TChl-*a* over the visible irradiance (400–700 nm) at a 1 nm spectral resolution using Equation (7). Comparison of the obtained A($\lambda$) and B($\lambda$) coefficients (Figure 6 and Table 2) with those revealed for a global ocean data set [44] (B95) shows the difference in B($\lambda$) at shorter wavelengths, while the A($\lambda$) is almost the same.

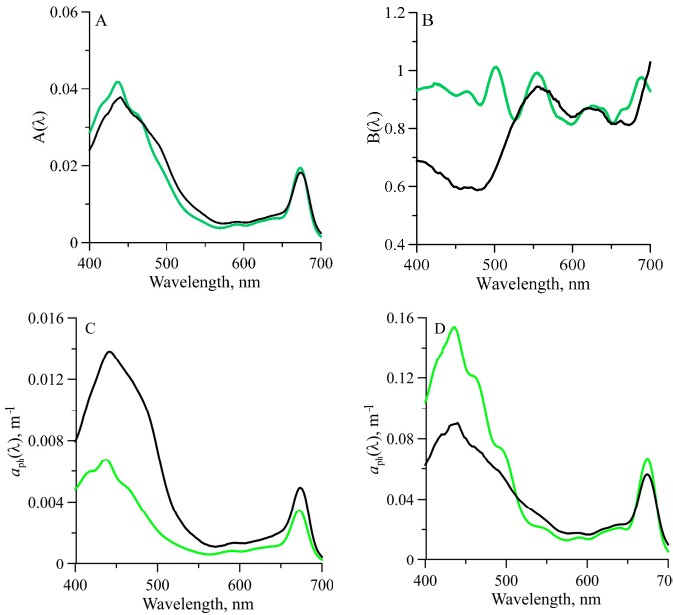

**Figure 6.** Values of the spectral constant $A(\lambda)$ (**A**) and $B(\lambda)$ (**B**) obtained when fitting power laws in the form of Equation (7) of the variations of phytoplankton light absorption coefficients ($a_{ph}(\lambda)$) versus the sum of chlorophyll *a* and phaeopigment concentrations (TChl-*a*) for the euphotic zone of the Atlantic sector of the Southern Ocean from 11 January–4 February 2020 (green lines). Comparison data are shown for a global ocean data set (black lines) described by [50]. The $a_{ph}(\lambda)$ spectra modeled using our parameterization coefficients when TChl-*a* equal 0.2 mg m$^{-3}$ (**C**) and 4.0 mg m$^{-3}$ (**D**) (green line) and [50] (black line) are shown for comparison.

Using the parameterization, the $a_{ph}(\lambda)$ spectra can be retrieved for a wide range of TChl-*a* variability in the euphotic zone. The $a_{ph}(\lambda)$ retrieved based on low (0.2 mg m$^{-3}$) and high (4.0 mg m$^{-3}$) TChl-*a* using the coefficients of our parameterization differed markedly from the $a_{ph}(\lambda)$ retrieved using the B95 parameterization. It is evident that $a_{ph}(\lambda)$ for low TChl-*a* is lower significantly than the B95, but for high TChl-*a* is higher than the B95 at all wavelengths (Figure 6). The apparent difference between the spectra is caused by the difference in B($\lambda$) coefficient.

**Table 2.** Spectral values of the constants (A($\lambda$) and B($\lambda$)) were obtained by fitting variations of $a_{\mathrm{ph}}(\lambda)$ versus the Chl-*a* plus phaeopigment concentrations (TChl-*a*) to power law as Equation (7).

| $\lambda$ | A($\lambda$) | B($\lambda$) | $\lambda$ | A($\lambda$) | B($\lambda$) |
|---|---|---|---|---|---|
| 400 | 0.0286 | 0.9319 | 478 | 0.0257 | 0.8877 |
| 402 | 0.0295 | 0.9347 | 480 | 0.0246 | 0.8822 |
| 404 | 0.0304 | 0.9356 | 482 | 0.0236 | 0.8803 |
| 406 | 0.0315 | 0.9371 | 484 | 0.0228 | 0.8859 |
| 408 | 0.0326 | 0.9396 | 486 | 0.0220 | 0.8960 |
| 410 | 0.0337 | 0.9407 | 488 | 0.0212 | 0.9127 |
| 412 | 0.0346 | 0.9410 | 490 | 0.0205 | 0.9316 |
| 414 | 0.0354 | 0.9420 | 492 | 0.0198 | 0.9529 |
| 416 | 0.0360 | 0.9425 | 494 | 0.0191 | 0.9723 |
| 418 | 0.0364 | 0.9440 | 496 | 0.0183 | 0.9896 |
| 420 | 0.0367 | 0.9450 | 498 | 0.0176 | 1.0027 |
| 422 | 0.0373 | 0.9551 | 500 | 0.0167 | 1.0103 |
| 424 | 0.0377 | 0.9545 | 502 | 0.0159 | 1.0122 |
| 426 | 0.0383 | 0.9530 | 504 | 0.0151 | 1.0071 |
| 428 | 0.0390 | 0.9531 | 506 | 0.0142 | 0.9961 |
| 430 | 0.0399 | 0.9505 | 508 | 0.0134 | 0.9807 |
| 432 | 0.0408 | 0.9471 | 510 | 0.0126 | 0.9599 |
| 434 | 0.0415 | 0.9433 | 512 | 0.0119 | 0.9382 |
| 436 | 0.0418 | 0.9399 | 514 | 0.0111 | 0.9130 |
| 438 | 0.0417 | 0.9344 | 516 | 0.0105 | 0.8905 |
| 440 | 0.0412 | 0.9304 | 518 | 0.0099 | 0.8694 |
| 442 | 0.0403 | 0.9249 | 520 | 0.0093 | 0.8534 |
| 444 | 0.0391 | 0.9221 | 522 | 0.0088 | 0.8415 |
| 446 | 0.0378 | 0.9191 | 524 | 0.0084 | 0.8330 |
| 448 | 0.0367 | 0.9166 | 526 | 0.0080 | 0.8296 |
| 450 | 0.0356 | 0.9136 | 528 | 0.0076 | 0.8314 |
| 452 | 0.0349 | 0.9141 | 530 | 0.0073 | 0.8377 |
| 454 | 0.0344 | 0.9154 | 532 | 0.0070 | 0.8470 |
| 456 | 0.0341 | 0.9179 | 534 | 0.0068 | 0.8617 |
| 458 | 0.0338 | 0.9222 | 536 | 0.0066 | 0.8787 |
| 460 | 0.0336 | 0.9242 | 538 | 0.0063 | 0.8943 |
| 462 | 0.0333 | 0.9262 | 540 | 0.0061 | 0.9133 |
| 464 | 0.0329 | 0.9274 | 542 | 0.0059 | 0.9309 |
| 466 | 0.0322 | 0.9270 | 544 | 0.0057 | 0.9451 |
| 468 | 0.0314 | 0.9242 | 546 | 0.0055 | 0.9599 |
| 470 | 0.0304 | 0.9196 | 548 | 0.0053 | 0.9706 |
| 472 | 0.0293 | 0.9134 | 550 | 0.0051 | 0.9802 |
| 474 | 0.0281 | 0.9047 | 552 | 0.0049 | 0.9886 |
| 476 | 0.0269 | 0.8957 | 554 | 0.0047 | 0.9925 |

| $\lambda$ | A($\lambda$) | B($\lambda$) | $\lambda$ | A($\lambda$) | B($\lambda$) |
|---|---|---|---|---|---|
| 556 | 0.0045 | 0.9909 | 630 | 0.0059 | 0.8759 |
| 558 | 0.0043 | 0.9832 | 632 | 0.006 | 0.8739 |
| 560 | 0.0041 | 0.9802 | 634 | 0.0061 | 0.8698 |
| 562 | 0.004 | 0.968 | 636 | 0.0062 | 0.8701 |
| 564 | 0.0039 | 0.9559 | 638 | 0.0063 | 0.8684 |
| 566 | 0.0039 | 0.9371 | 640 | 0.0063 | 0.8685 |
| 568 | 0.0038 | 0.9233 | 642 | 0.0063 | 0.8621 |
| 570 | 0.0038 | 0.9064 | 644 | 0.0063 | 0.8533 |
| 572 | 0.0039 | 0.8898 | 646 | 0.0063 | 0.8409 |
| 574 | 0.0039 | 0.8744 | 648 | 0.0064 | 0.8282 |
| 576 | 0.004 | 0.8603 | 650 | 0.0065 | 0.819 |
| 578 | 0.0041 | 0.8498 | 652 | 0.0068 | 0.8172 |

**Table 2.** *Cont.*

| λ | A(λ) | B(λ) | λ | A(λ) | B(λ) |
|---|---|---|---|---|---|
| 580 | 0.0042 | 0.8411 | 654 | 0.0074 | 0.8217 |
| 582 | 0.0043 | 0.8394 | 656 | 0.0082 | 0.8305 |
| 584 | 0.0045 | 0.8355 | 658 | 0.0094 | 0.8416 |
| 586 | 0.0046 | 0.8346 | 660 | 0.0108 | 0.8514 |
| 588 | 0.0047 | 0.8307 | 662 | 0.0124 | 0.8612 |
| 590 | 0.0047 | 0.8258 | 664 | 0.0142 | 0.8649 |
| 592 | 0.0047 | 0.8229 | 666 | 0.0159 | 0.867 |
| 594 | 0.0047 | 0.8174 | 668 | 0.0175 | 0.8667 |
| 596 | 0.0046 | 0.8163 | 670 | 0.0186 | 0.869 |
| 598 | 0.0045 | 0.8135 | 672 | 0.0193 | 0.874 |
| 600 | 0.0045 | 0.8148 | 674 | 0.0194 | 0.8845 |
| 602 | 0.0044 | 0.8182 | 676 | 0.0189 | 0.8991 |
| 604 | 0.0044 | 0.8227 | 678 | 0.0177 | 0.9163 |
| 606 | 0.0045 | 0.8302 | 680 | 0.0159 | 0.935 |
| 608 | 0.0046 | 0.8398 | 682 | 0.0138 | 0.9501 |
| 610 | 0.0047 | 0.8494 | 684 | 0.0114 | 0.9656 |
| 612 | 0.0049 | 0.8552 | 686 | 0.0092 | 0.9721 |
| 614 | 0.0051 | 0.8635 | 688 | 0.0072 | 0.9745 |
| 616 | 0.0052 | 0.8664 | 690 | 0.0056 | 0.975 |
| 618 | 0.0053 | 0.8696 | 692 | 0.0042 | 0.9692 |
| 620 | 0.0055 | 0.8705 | 694 | 0.0033 | 0.9612 |
| 624 | 0.0056 | 0.8777 | 696 | 0.0025 | 0.9491 |
| 622 | 0.0056 | 0.8737 | 698 | 0.002 | 0.9383 |
| 626 | 0.0057 | 0.8775 | 700 | 0.0016 | 0.8634 |
| 628 | 0.0058 | 0.878 | | | |

### 3.5. Light Absorption by NAP, CDOM, and CDM

The $a_{NAP}(438)$ varied in the range of 0.002–0.033 m$^{-1}$. Values of $a_{NAP}(438)$ showed a positive trend with TChl-*a* but with only a slight correlation (r$^2$ = 0.42) (Figure 7). The share of the NAP in total particulate absorption was variable (Figure 4). The ratio between $a_{NAP}(438)$ and $a_p(438)$ was on average 0.24 ± 0.12 in the surface layer, 0.15 ± 0.08 in the euphotic zone, and 0.30 ± 0.13 below 1% PAR depth (Figure 8). The S$_{NAP}$ values were found to vary between 0.006 to 0.016 nm$^{-1}$ with an average value of 0.011 ± 0.0017 nm$^{-1}$.

Light absorption by CDOM ($a_{CDOM}(438)$) changed by almost two orders of magnitude (0.0054–0.19 m$^{-1}$), but no dependence of the $a_{CDOM}(438)$ on TChl-*a* was observed for neither the surface, the photosynthesis zone, nor the deeper layer (Figure 7). It should be noted that the variability of CDOM absorption at 438 nm was comparable with that of phytoplankton absorption (about two orders of magnitude) and significantly higher than NAP absorption variability. S$_{CDOM}$ values were found to vary between 0.0082 and 0.029 nm$^{-1}$ with an average value of 0.013 ± 0.0034 nm$^{-1}$ (Figure 4). At the surface, the S$_{CDOM}$ values co-varied with photosynthetic available radiation incident on the surface (Figure 9). Based on all datasets, an inverse relation between S$_{CDOM}$ and $a_{CDOM}(438)$ was revealed (Figure 9). The relationship was described well (r$^2$ = 0.60) by the following equation:

$$S_{CDOM} = 0.0054 \times a_{CDOM}(438)^{-0.27} \tag{12}$$

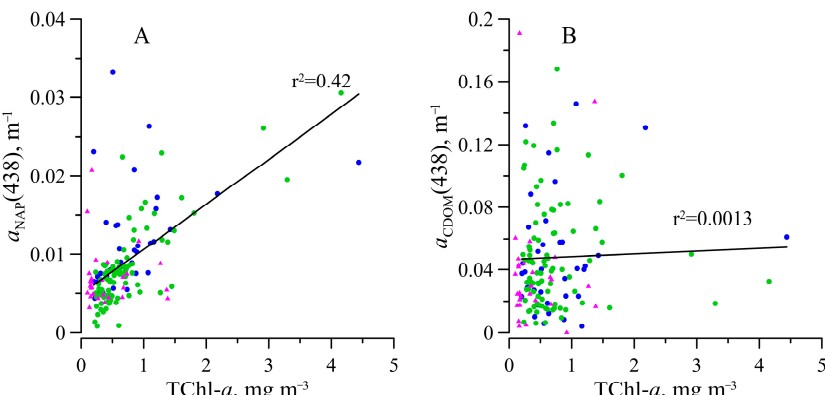

**Figure 7.** Effect of the sum of chlorophyll *a* concentration and phaeopigments (TChl-*a*) on light absorption coefficients of non-algal particles ($a_{NAP}(438)$) (**A**) and coloured dissolved organic matter at 438 nm ($a_{CDOM}(438)$ ) (**B**). Symbol notations are the same as in Figure 5.

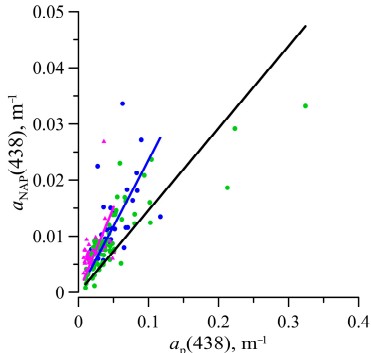

**Figure 8.** Relationship between light absorption coefficient of non-algal particles ($a_{NAP}(438)$) and total particulate absorption ($a_p(\lambda)$) at 438 nm. Symbols are denoted as in Figure 5. Blue, black and purple lines denote the linear relationship for the surface ($a_{NAP}(438) = 0.24 \times a_p(438)$ $r^2 = 0.83$ (n = 35)), for euphotic ($a_{NAP}(438) = 0.15 \times a_p(438)$, $r^2 = 0.75$ (n = 127)) and below layers ($a_{NAP}(438) = 0.30 \times a_p(438)$, $r^2 = 0.72$ (n = 28)), respectively.

No dependence of CDOM absorption on water salinity was revealed ($r^2 = 0.040$) (Figure 10).

The light absorption coefficient of coloured detrital mater ($a_{CDM}(\lambda)$), which is the sum of NAP and CDOM absorption coefficients, varied by about one order of magnitude. Values of $a_{CDM}(438)$) was in the range of 0.010–0.20 m$^{-1}$ with an average value of 0.055 m$^{-1}$. CDM spectral slope coefficient ($S_{CDM}$) values varied about three times (0.0061–0.021 nm$^{-1}$), mean was equal to 0.012 ± 0.0025 nm$^{-1}$. An inverse relationship between the $a_{CDM}(438)$ and $S_{CDM}$ was revealed (Figure 9). This link was described by the power function (Equation (13)), but the determination coefficient was less ($r^2 = 0.44$) than that for CDOM (Equation (12)). The reason is the NAP absorption, which is not co-varied with CDOM absorption.

$$S_{CDM} = 0.0062 \times a_{CDM}(438)^{-0.22}.$$ (13)

### 3.6. Absorption Budget

To analyse the contributions of phytoplankton, NAP, and CDOM to the total non-water absorption, their normalized coefficients were shown on a ternary plot at 438 and 490 nm that corresponded to the specific characteristic of phytoplankton, in particular maximum absorption (438 nm), as well as to the waveband of most ocean colour sensors (490 nm) (Figure 11).

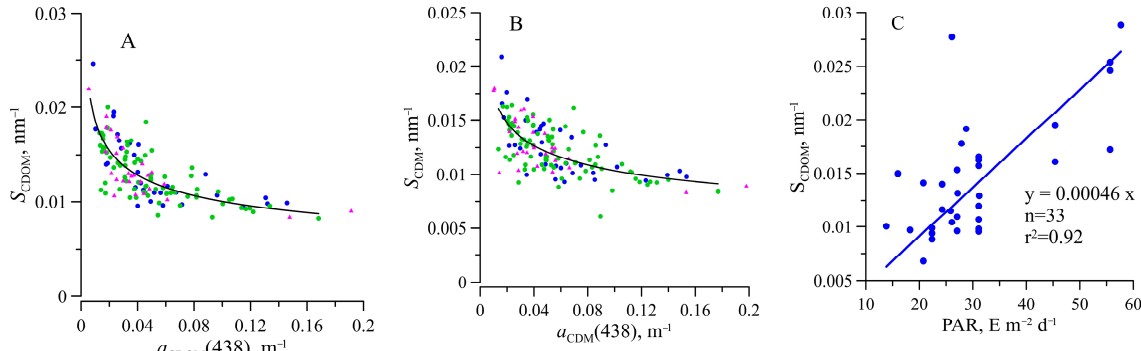

**Figure 9.** Dependence of the spectral slope coefficient ($S_{CDOM}$ and $S_{CDM}$) on light absorption coefficient at 438 nm of coloured dissolved organic matter ($a_{CDOM}(438)$) (**A**) coloured detrital matter ($a_{CDM}(438)$)) (**B**) and the effect of daily photosynthetic available radiance (PAR) on the $S_{CDOM}$ (**C**). Symbol notations are the same as in Figure 5. The black line denotes the relationship for the euphotic zone: $S_{CDOM} = 0.0054 \times a_{CDOM}(438)^{-0.27}$, n = 105, r$^2$ = 0.60; $S_{CDM} = 0.0054 \times a_{CDOM}(438)^{-0.27}$ n = 105, r$^2$ = 0.41, the blue line stands for the surface $S_{CDOM} = 0.00046 \times PAR$, r$^2$ = 0.92.

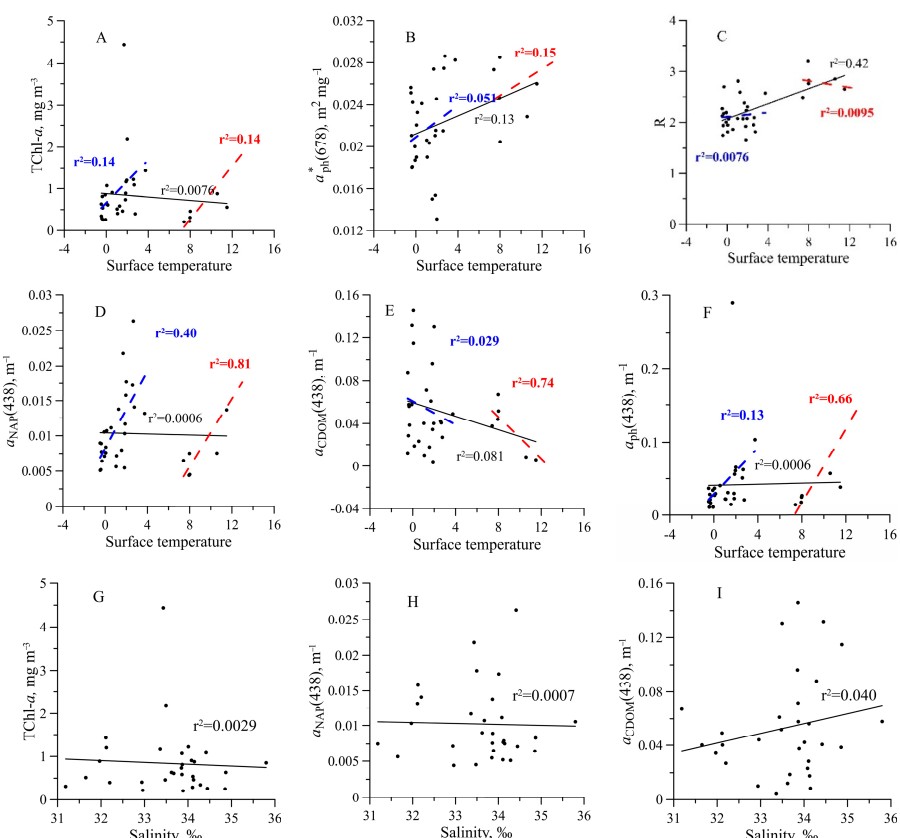

**Figure 10.** Influence of the surface temperature (**A–F**) and salinity (**G–I**) on the sum of chlorophyll *a* and phaeopigments concentration (TChl-*a*) (**A,G**), TChl-*a* specific light absorption coefficient of phytoplankton at 678 nm ($a_{ph}^*(678)$) (**B**), the ratio between phytoplankton light absorption coefficients at blue and red peaks (R) (**C**), non-algal particles light absorption coefficient at 438 nm ($a_{NAP}(438)$) (**D,H**), coloured dissolved organic matter light absorption coefficient at 438 nm ($a_{CDOM}(438)$) (**E,I**) and light absorption coefficient of phytoplankton at 438 nm ($a_{ph}(438)$) (**F**) in the surface layer of the Southern Ocean.

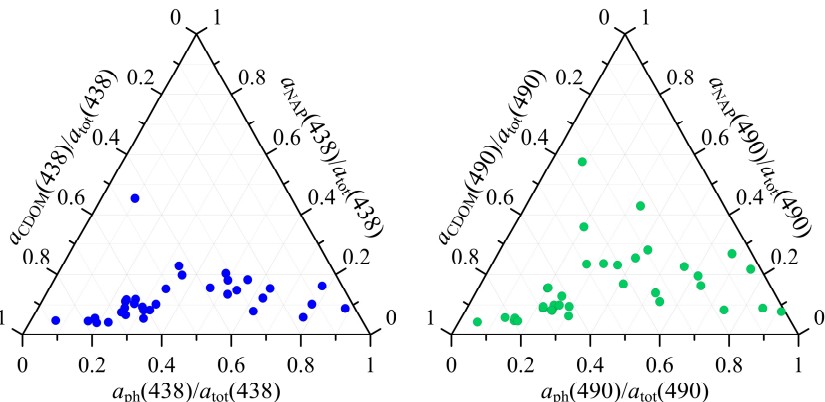

**Figure 11.** Contribution of phytoplankton ($a_{ph}(\lambda)/a_{tot}(\lambda)$), non-algal particles ($a_{NAP}(\lambda)/a_{tot}(\lambda)$ ), and coloured dissolved organic matter ($a_{CDOM}(\lambda)/a_{tot}(\lambda)$ ), to the total non-water absorption at 438 nm (blue symbols) and at 490 nm (green symbols) in the surface layer of the Southern Ocean from 11 January–4 February 2020.

In the surface layer and wavelengths examined, $a_{CDOM}(\lambda)$ or $a_{ph}(\lambda)$ dominated the total non-water absorption. An increase in the phytoplankton (or CDOM) share in the total non-water absorption was due to an increase in phytoplankton absorption coefficient (and TChl-*a*). In this case, the CDOM share decreased, correlating with the CDOM absorption coefficient (Figure 12). The phytoplankton and CDOM shares were highly variable. At 438 nm, the CDOM contribution varied from 3 to 92% with an average value of 45 ± 26%; phytoplankton contribution varied from 6 to 87% with an average of 42 ± 21%. The NAP contribution, with the exception of one point, was the smallest 2–21% with an average value equal to 13 ± 8%. At 490 nm CDOM and phytoplankton remained dominant in the total non-water absorption. However, the share of NAP increased up to 42% in comparison with that observed at 438 nm (Figure 11).

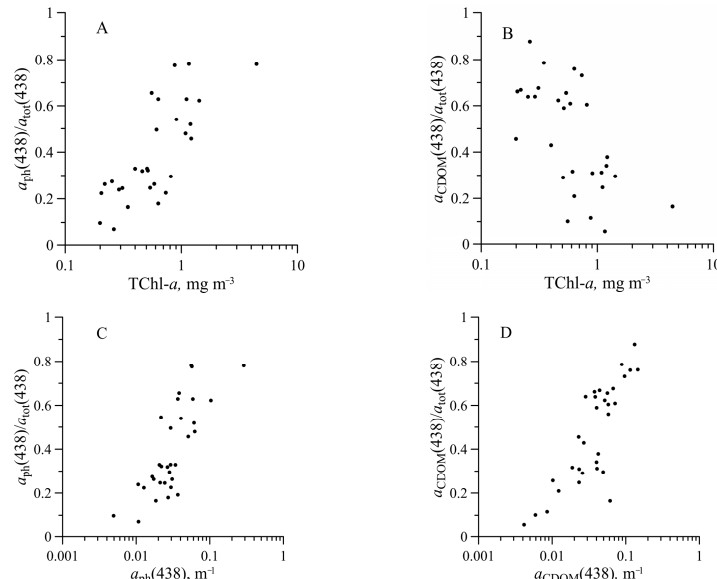

**Figure 12.** Relationship of the contribution of phytoplankton ($a_{ph}(438)/a_{tot}(438)$) (**A,C**) and coloured dissolved organic matter ($a_{CDOM}(438)/a_{tot}(438)$ ) (**B,D**) to the total non-water absorption at 438 nm with the sum of chlorophyll *a* and phaeopigments concentration (TChl-*a*) (**A,B**) and with $a_{ph}(438)$ (**C**) and with $a_{CDOM}(438)$ (**D**).

## 4. Discussion

The study area was very contrasting in terms of hydrological characteristics. The currents with relatively warm (FC and DP) and cold waters (in the BS from the Weddell Sea), as well as frontal zones (in the BS and the northern PB), caused different environmental factors for phytoplankton growth regarding not only temperature but also the light intensity and nutrient availability [55,56]. As a result, the TChl-*a* varied widely across investigated regions. The TChl-*a* range of variability and vertical fluorescence profiles were in good agreement with the results obtained in the western-northern Antarctic Peninsula in austral summer [26,33,57,58]. Although in some water areas, the influence of temperature on TChl-*a* was noted (Figure 10) but generally, for the entire study area, there was no dependence of TChl-*a* on temperature. It is due to the multifactorial dependence of TChl-*a*. TChl-*a* depends on phytoplankton photosynthesis/growth, which is determined not only by temperature but also by the light "history" of phytoplankton related to water stratification and by nutrient availability as well [14,28,59,60]. The Southern Ocean is known to be a high-nutrient ecosystem [12], but iron limitation observed locally resulted in the heterogeneity of the region in terms of phytoplankton photophysiology and photosynthetic production [17,61,62]. Moreover, zooplankton grazing affects phytoplankton biomass and, consequently, the TChl-*a* [63]. Hence, the observed TChl-*a* variability is due to a combination of environmental factors.

However, for such a wide range of TChl-*a* variability, the relative stability of $a^*_{ph}(678)$, $a^*_{ph}(438)$ and R was noted within the euphotic zone. Based on known regularities of $a^*_{ph}(\lambda)$ variability [64], stability of $a^*_{ph}(\lambda)$ indicates that pigment package effect associated with intracellular pigment concentration and phytoplankton species and the cell size composition changed insignificantly with depth within the euphotic zone, if changed at all. Such homogeneity in the vertical distribution of functional characteristics of phytoplankton (in particular, specific light absorption) was probably due to the insufficient stability of the water column (although temperature and salinity changed with depth), which led to vertical mixing and the planktonic algae could not manage to change their light-absorbing characteristics with depth [65].

It could be noted that below the euphotic zone a tendency to decrease in $a^*_{ph}(678)$, $a^*_{ph}(438)$ and R was observed (Figure 4). Below the euphotic zone, phaeopigment share summed with chlorophyll *a* was increased. In most stations, phaeopigments share in the euphotic zone was 4–35 % except for three stations with 58–75 %, where TChl-*a* exceeded 1 mg m$^{-3}$. The high content of phaeopigments (58–75%) in the Southern Ocean was detected before and was linked to increased grazing pressure [32]. An increase in the phaeopigment share below the euphotic zone resulted in a decrease in $a^*_{ph}(\lambda)$ and R (Figure 4). This effect is related to the phaeopigment-specific light absorption coefficient (absorbance extinction), which is almost three times lower than that of chlorophyll *a* [66].

Parameterization of the phytoplankton absorption within the euphotic zone revealed a noticeably higher coefficient B($\lambda$) for the blue spectrum domain compared to what was generally reported in the literature [26,50,52]. The observed invariance of $a^*_{ph}(438)$ and $a^*_{ph}(678)$ (while TChl-*a* changed by 20 times), which resulted in a coefficient B($\lambda$) close to 1, explains no increase in intracellular pigment package degree [67,68] under TChl-*a* rise in contrast to results observed before in different regions of the global ocean [50,52] and the Southern Ocean [26,28]. General pattern of $a^*_{ph}(\lambda)$-TChl-*a* relationship is a decrease in $a^*_{ph}(\lambda)$ with increasing TChl-*a* [26,28,32,50,52].

Another pattern of $a^*_{ph}(\lambda)$-TChl-*a* relationship observed in our research could be due to the fact that in the summer of 2020, increasing TChl-*a* was likely associated with changes in the size structure of phytoplankton and, in particular, associated with a shift towards small-celled phytoplankton observed during our scientific cruise [69] and in this region before [70]. This change in the cell size explains the observed absence of a decrease in TChl-*a* specific absorption, which was expressed in the phytoplankton absorption parameterization by the B coefficient near 1. In the Southern Ocean, a shift in dominant phytoplankton taxon has

been noted: from large diatoms to small dinoflagellates, including cryptophyte algae [1]. Our measurements of $a_{ph}(\lambda)$ showed that local maxima at 490 and 550 nm were noticeable at particular stations (Figure 13).

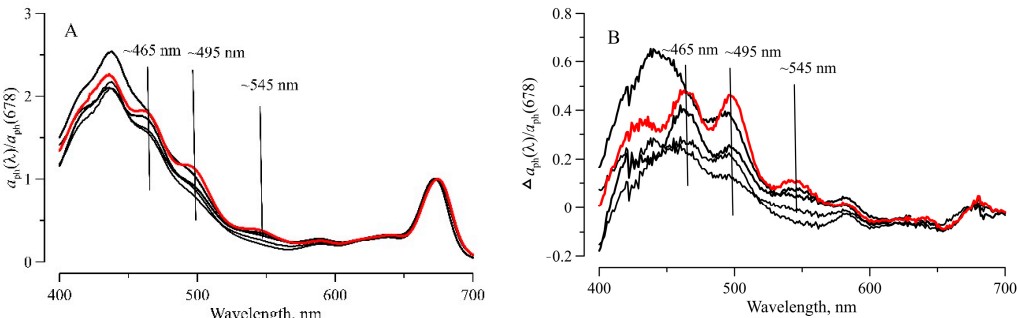

**Figure 13.** Phytoplankton light absorption spectra ($a_{ph}(\lambda)$) normalized on its value at red maximum ($a_{ph}(678)$) at stations (6585, 6609, 6614, 6631, RS4) (**A**); the difference between spectra $a_{ph}(\lambda)/a_{ph}(678)$ with and without local maxima (**B**). The red line denotes the spectrum with a more pronounced local maximum at station 6609.

These local maxima are associated with absorption bands of phycoerythrin, which contains in the pigment complex of cryptophyte and cyanobacteria [53,71,72]. Previously, we showed that local maxima became visible on the spectrum if a contribution of the phycoerythrin-containing species to the total phytoplankton biomass reached ca. 20% [73]. At a smaller phycoerythrin share, the local maxima were masked by other pigment absorption. The spectral features of $a_{ph}(\lambda)$ confirm the assumption of the predominant development of small-celled species (including cryptophytes) in more trophic waters, which may be the reason for the noted features in the parametrization of light absorption by phytoplankton (B coefficient near 1 in the blue domain) obtained in our research. Increased abundance of the cryptophytes (assessed indirectly via $a_{ph}(\lambda)$ shape) in more trophic and warmer waters agrees with earlier results that showed that the cryptophytes were most abundant during the summer months, and they were associated with warmer surface waters [70,74].

Generally, the NAP light absorption is determined by the absorption of suspended organic (non-living) particles, which are not associated with phytoplankton (bacteria and detritus), and absorption of mineral particles which are typical for shallow and coastal waters [75,76]. The obtained variable NAP absorption contribution to $a_p(438)$ (Figure 7) and a slight correlation between $a_{NAP}(438)$ and TChl-a ($a_{ph}(438)$) in the euphotic zone (Figure 4) are in good agreement with that observed earlier [32,33]. Salinity did not affect $a_{NAP}(438)$ variability (Figure 10). Consequently, in the investigated area, an influence of ice melting on NAP was not detected. NAP is likely to have both a biological and physical origin, which could be related to the currents and frontal zones in the area studied (Figure 1). The $S_{NAP}$ ($0.011 \pm 0.0017$ nm$^{-1}$) agreed with results obtained in different regions of the global ocean [33,35,52,77,78]. The $S_{NAP}$ values were less variable as compared to $S_{CDOM}$, which was consistent with the results of ref. [52]. The obtained $S_{CDOM}$ values agreed with data in the western-northern Antarctic Peninsula [33]. The range of $S_{CDOM}$ variability obtained for the Southern Ocean corresponded to that obtained for the open ocean (0.015–0.030 nm$^{-1}$) [79] and the European coastal waters 0.014–0.020 nm$^{-1}$ [52], but with a lower limit to the range.

The $S_{CDOM}$ showed an inverse relation with $a_{CDOM}(438)$ as observed elsewhere [35,79]. The positive correlation between the $S_{CDOM}$ and PAR incident on the surface was revealed (Figure 9). It justifies that the inverse trend of $S_{CDOM}$ against $a_{CDOM}(438)$ is related to photobleaching near the surface. Due to CDOM photodegradation, low-molecular-weight compounds occurred, which resulted in $S_{CDOM}$ increasing and $a_{CDOM}(438)$ decreasing [80].

CDOM absorption at 438 nm was found to vary widely ((0.0054–0.19 m$^{-1}$). The high CDOM variability was observed before in the Antarctic Peninsula area [81]. No correlation between CDOM absorption and both TChl-*a* and salinity (Figure 7, 10) suggests the influence of various factors on the CDOM content: biological (phytoplankton, bacteria, and other pelagic organisms) and physical (photobleaching, ice melting), which can operate in different directions [81–83].

We obtained that at the surface layer and examined wavelengths (438 and 490 nm), CDOM and phytoplankton alternately dominated the total non-water absorption (Figure 11). Earlier investigations showed only the phytoplankton domination in the northern Antarctic Peninsula in 2013–2014 [33] and the Indian sector of the Southern Ocean [28]. In our research, the increasing share of phytoplankton or CDOM resulted from an increase in the absolute content of this optically active component (Figure 12). In the more trophic area, the CDOM content was lower than in less trophic waters. The region of our studies was highly contrasted in terms of TChl-*a* content in the surface layer and in terms of hydrological and optical characteristics of the waters as well. As a consequence, in our studies, there were areas where phytoplankton or CDOM dominated in light absorption in the upper layer (Figure 14). The increase in the share of phytoplankton or CDOM (Figure 14) was, in general, coincident with the rise of their content and absorption coefficient.

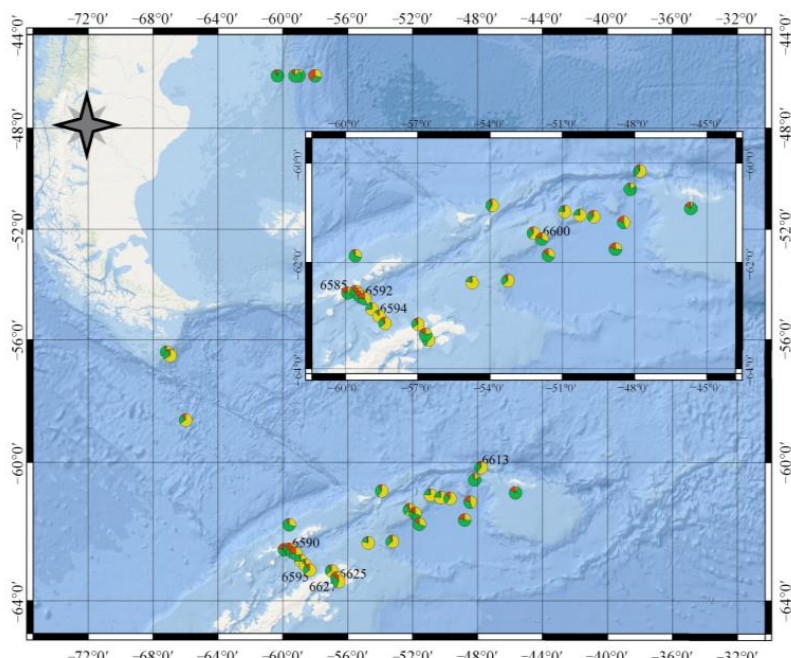

**Figure 14.** Contribution of the phytoplankton (green), non-algal particles (brown), and coloured dissolved organic matter (yellow) to the total non-water absorption at 438 nm in the surface layer of the Southern Ocean from 11 January–4 February 2020.

Comparison of the Chl-*a* measured in this research with satellite retrievals shows that satellite assessments underestimate (except two stations of the FC: the first corresponding to 1:1 line and the second corresponding to above 1:1 line) the Chl-*a* about two times (Figure 15, Table 3), which agreed with the previous studies [20–25]. The satellite Chl-*a* validation in ref. [24] fulfilled on a larger dataset than in our research showed a trend to an increase in deviation of the satellite data from the 1:1 correspondence with an increase in Chl-*a* (Figures 2 and 3 in [24]). This trend is possibly related to the high variability in CDOM absorption without correlation with TChl-*a*, which leads to a decrease in the CDOM share in total non-water absorption at high TChl-*a*, as observed in our research (Figure 12).

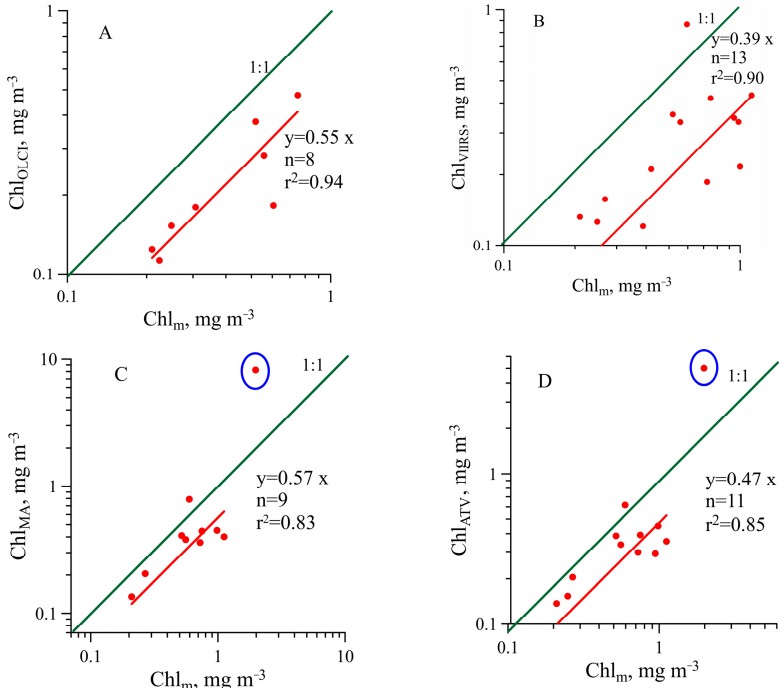

**Figure 15.** Comparison of the satellite chlorophyll *a* concentration Chl and the concentration measured in situ ($Chl_m$). The green line 1:1 represents an ideal correspondence: (**A**)—S3A-OLCI data ($Chl_{OLCI}$), (**B**)—SNPP-VIIRS data ($Chl_{VIIRS}$), (**C**)—Aqua-MODIS data ($Chl_{MA}$) and (**D**)—Merged_ATV data ($Chl_{ATV}$). The red line represents the correlation between $Chl_m$ and satellite data without one maximal satellite value—dot in the blue circle (**C,D**).

**Table 3.** Validation of satellite (Merged ATV, Aqua-MODIS, S3A-OLCI, SNPP-VIIRS) chlorophyll *a* retrieval based on statistical metrics: bias (Equation (3)) and MAE (Equation (4)).

| Satellite Scanners | n | bias | MAE |
|:---:|:---:|:---:|:---:|
| Merged ATV | 12 | 0.63 | 1.9 |
| Aqua-MODIS | 17 | 0.51 | 2.4 |
| S3A-OLCI | 8 | 0.54 | 1.8 |
| SNPP-VIIRS | 14 | 0.47 | 2.2 |
| without maximum value (one point highlighted in Figure 15) | | | |
| Merged ATV | 11 | 0.55 | 1.8 |
| Aqua-MODIS | 16 | 0.45 | 2.3 |

## 5. Conclusions

Expedition (No. 79) on RV "Akademik Mstislav Keldysh" allowed us to collect the dataset to investigate the variability in spectral light absorption coefficients by all optically active components in the Atlantic region of the Southern Ocean in the austral summer of 2020. The high spatial variability in $a_{CDOM}(\lambda)$ (uncorrelated with TChl-*a*) and CDOM share in the total non-water absorption resulting in a shift in the dominance from CDOM to phytoplankton reflects the regional specificity of the bio-optical properties of the Southern Ocean that cause deviations (underestimation about two times) in the global bio-optical algorithms (different scanners) when applied to the Southern Ocean (Figure 15, Table 3). The performed parametrization of the link between phytoplankton absorption coefficients and TChl-*a* within the visible domain in 1 nm increments allows the retrieval of a spectrum based on TChl-*a* data required for the assessment of the photosynthetic characteristics of phytoplankton and primary production, as well as to solve the inverse task, i.e., the assessment of the TChl-*a* based on remote sensing data using the developed three-band algorithm for separating light absorption by phytoplankton and coloured detrital matter [34].

The results of NAP, CDOM, and CDM light absorption parameterization and the retrieved relationship between $S_{CDOM}$ ($S_{CDM}$) and CDOM (CDM) absorption coefficient can be useful for adaptation of the three-band algorithm developed for the Black Sea to the Southern Ocean waters, which is planned to be realized in future research.

**Author Contributions:** Conceptualization, T.C.; methodology, T.C. and N.M.; software, A.B.; formal analysis, T.C., N.M., T.E. and E.S.; resources, N.M., V.A. and P.S.; writing—original draft preparation, T.C.; writing—review and editing, T.C. and P.S.; visualization, N.M.; supervision, T.C.; project administration, T.E.; funding acquisition, T.C. All authors have read and agreed to the published version of the manuscript.

**Funding:** Analysis of variability of the spectral light absorption by optically active components and manuscript writing were supported by Russian Science Foundation (grant No. 22-27-00790).

**Data Availability Statement:** Not applicable.

**Acknowledgments:** We would like to thank the captains and crew of the RV "Akademik Mstislav Keldysh" for technical assistance. The authors are indebted to NASA's Ocean Biology Processing Group (OBPG) for providing access to the satellite ocean colour dataset used in this work. The authors are very thankful to anonymous reviewers for their comments and helpful suggestions.

**Conflicts of Interest:** The authors declare no conflict of interest. The funders had no role in the design of the study; in the collection, analyses, or interpretation of data; in the writing of the manuscript, or in the decision to publish the results.

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
