# Peer review of "Parameterization of Light Absorption of Phytoplankton, Non-Algal Particles and Coloured Dissolved Organic Matter in the Atlantic Region of the Southern Ocean (Austral Summer of 2020)"

_remotesensing, doi:10.3390/rs15030634_

Round 1

Reviewer 1 Report

1. Recommendation:

Minor Revision

2. Overview and general recommendation:

The manuscript by Churilova et al. presents an analysis of light absorption of phytoplankton, non-algal particles and colored dissolved organic matter in the Atlantic region. The authors parametrized light absorption of phytoplankton (for visible domain with 1 nm step), NAP and CDOM based on newly collected in situ data in austral summer in the Atlantic region of the Southern Ocean. The current topic is really of interest and it’s the basis for the separation of organic and inorganic particulate matter from an optical point of view. In general, the document is carefully organized, the introduction, materials and applied methodology are well presented, the manuscript is well written with good English language and style, and the references are appropriated. The results and their management are convincing, and the figures support the results and discussion properly. The decision of this review is that this manuscript can be accepted for publication after revising the following minor comments.

3. Minor comments:

(1) When using abbreviations for the first time, it is recommended to add the full name, especially if it is uncommon or self-defined, for example, SCDOM in Line 29.

(2) Line 226-230: The range of values should be connected by -, rather than ….

(3) Please ensure the uniformity of station expressions in the whole text (multiple expressions appear in the whole text, such as st6576, st.6576, station6576), if abbreviation is used then they should be uniformly expressed as st***, if not then they should be all expressed as station***.

(4) Figure 2: Much higher resolution pictures are recommended.

(5) Quality control (such as median filtering) need to be done for all your optical profiles, for example there are lots of spikes along the profiles of Chla at figure2 which may due to possible signal system bias. So, detail description of quality control for optical profiles need to be added in Section 2, and Figure 2 need to be revised.

(6) Figure 3: Legend of each color line need to be added.

(7) Line 289: reference is needed for “Fisher's and Student's criteria”.

(8) Figure 4 and Figure 9: the decimal point of the coordinate scale values should be ‘.’ rather than ‘,’.

(9) Please explain why the fitted parameters B(438nm) and B(678nm) in Table1 are different with those in Table2.

(10) Figure 15: Legend of each line need to be added. Moreover, there are no black line in Figure 15 and what are the red lines mean, please explained.

(11) Providing a concept map will be more helpful for all readers to sort out the lineage of your article and the application of your article. If some figures can be provided to classify the contribution/effect of phytoplankton/particle sizes, package effect, water stratification on the aph* will be perfect, although it's really difficult.

Author Response

Response to Reviewer 1

The authors thank the reviewer for an attentive reading of the manuscript, made comments and useful recommendations, which made it possible to improve the manuscript significantly.

comments:

(1) When using abbreviations for the first time, it is recommended to add the full name, especially if it is uncommon or self-defined, for example, SCDOM in Line 29.

Response Thanks. It’s been done. “Relationships between spectral slope coefficient (SCDOM/SCDM) and CDOM (CDM) absorption were revealed.”

(2) Line 226-230: The range of values should be connected by -, rather than ….

Response Thanks. It’s been changed

(3) Please ensure the uniformity of station expressions in the whole text (multiple expressions appear in the whole text, such as st6576, st.6576, station6576), if abbreviation is used then they should be uniformly expressed as st***, if not then they should be all expressed as station***.

Response Thanks. It’s been corrected – expressed as “station”

(4) Figure 2: Much higher resolution pictures are recommended.

Response Thanks.  It’s been done

(5) Quality control (such as median filtering) need to be done for all your optical profiles, for example there are lots of spikes along the profiles of Chla at figure2 which may due to possible signal system bias. So, detail description of quality control for optical profiles need to be added in Section 2, and Figure 2 need to be revised.

Response Thank you very much for this recommendation.  But we believe that such a “noise” of data does not prevent us from solving the scientific tasks of this particular research. However, your advice is undoubtedly very useful and will be used if necessary

(6) Figure 3: Legend of each color line need to be added.

Response Thanks. It’s been done

Figure 3. Spectral light absorption coefficient of particles (, blue line), phytoplankton (, green line), non-algal particles (, brown line) and coloured dissolved organic matter (, yellow line) at particular stations corresponding to all regions investigated in the Southern Ocean.

(7) Line 289: reference is needed for “Fisher's and Student's criteria”.

Response Thanks. It’s been added

McDonald J.H. Handbook of Biolological Statistics. Third Edition. Baltimore, Maryland, U.S.A: Sparky House Publishing; 2014. University of Delaware.

(8) Figure 4 and Figure 9: the decimal point of the coordinate scale values should be ‘.’ rather than ‘,’.

Response Thanks. It’s been done (Figures 3,4,6,7,9,11,12)

(9) Please explain why the fitted parameters B(438nm) and B(678nm) in Table1 are different with those in Table2.

Response Many thanks.  It was a mistake (in the table 1). It’s been corrected

(10) Figure 15: Legend of each line need to be added. Moreover, there are no black line in Figure 15 and what are the red lines mean, please explained.

Response Thanks.  The legend has been corrected. (Before submission of the manuscript black line was replaced by red line, but the legend was not corrected. But now everything is correct ).

Figure 15. Comparison of the satellite chlorophyll a concentration Chl and the concentration measured in situ (Chlm). The green line 1:1 represents an ideal correspondence: A ‒ S3A-OLCI data (ChlOLCI), B – SNPP-VIIRS data (ChlVIIRS), C – Aqua-MODIS data (ChlMA) and D – Merged_ATV data (ChlATV). Red line represents correlation between Chlm and satellite data without one maximal satellite value - dot in blue circle (C, D).

(11) Providing a concept map will be more helpful for all readers to sort out the lineage of your article and the application of your article. If some figures can be provided to classify the contribution/effect of phytoplankton/particle sizes, package effect, water stratification on the aph* will be perfect, although it's really difficult.

Response Many thanks for interesting idea. We will try to realized it in the next research.

Best regards

Author team

Reviewer 2 Report

The major comments are included in the manuscript. The authors has attempted to bring novelty in the research. However, it is disappointing to mention that, the authors has referred limited publications and not referred  studies from the Indian Sector of Southern Ocean. The corrections and clarifications can be made and manuscript can be modified. Also, the values (especially the minimum, maximum and average has to be rechecked). All the best.

Author Response

Response to Reviewer 2

The authors are very thankful to the reviewer for a careful reading of the manuscript.

Comments and recommendations done by the reviewer, are very useful for an improvement of the manuscript. The manuscript was revised 

Full reply you can find in the attached file

Best regards

Author team
